# Honey bees bred for *Varroa* sensitive hygiene trait demonstrate resistance to chalkbrood disease

Isabell Dyrbye-Wright[1]*, Michael Simone-Finstrom[2], Elizabeth M. Walsh[2], Marla Spivak[1]

1 Department of Entomology, University of Minnesota, St. Paul, Minnesota, United States of America,
2 USDA-ARS, Honey Bee Breeding, Genetics, and Physiology Research, Baton Rouge, Louisiana, United States of America

* dyrby003@umn.edu

## Abstract

To improve health and vitality of honey bees (*Apis mellifera*) beekeepers can propagate stocks that demonstrate resistance to both parasites and pathogens. Most breeding programs focus on resistance to *Varroa destructor* mites and/or brood pathogens. Colonies bred specifically for the trait *Varroa* Sensitive Hygiene (VSH), exhibit a high level of resistance to the parasitic mites. Still, they have never been explicitly tested for resistance to brood diseases. The goal of this study was to test if colonies bred for VSH are both mite and disease resistant. Over two years (2023 and 2024) and in two locations (University of Minnesota and the USDA-ARS lab in Baton Rouge, Louisiana), we compared colonies from the Pol-line bred specifically for VSH to colonies from a commercial source. The Commercial colonies in this study were not selected specifically for *Varroa* resistance but were selected for "general" hygienic behavior using the freeze-killed brood (FKB) assay. We challenged colonies within each line with *Ascosphara apis,* a fungus that causes chalkbrood, and quantified mites, disease and hygienic behavior. Our study demonstrated that bees from the Pol-line bred for VSH are just as resistant to chalkbrood as bees from a commercial line bred for hygienic behavior. Results confirmed that the Pol-line was more mite resistant than the Commercial, as it had significantly lower mites in two of three trials. Both the Pol-line and Commercial colonies had high levels of hygienic behavior. These results indicate that VSH-selected honey bees respond to both mite-infested and disease-infected brood. Further comparative studies are needed to clarify any differences in genetic mechanisms and olfactory sensitivity mediating the VSH-trait and general hygienic behavior. On a practical level, using honey bees selected for VSH in beekeeping operations could help curb losses, improve honey bee health, and reduce financial burdens caused by *Varroa* and diseases.

the Creative Commons CC0 public domain dedication.

**Data availability statement:** All R script and CSV files used in this study are publicly available from the Dryad data base (https://doi.org/10.5061/dryad.v6wwpzh6p) Dyrbye-Wright, Isabell; Walsh, Elizabeth M; Simone-Finstrom, Michael; Spivak, Marla (2025). Bees bred for Varroa sensitive hygiene trait demonstrate resistance to chalkbrood disease [Dataset]. Dryad. https://doi.org/10.5061/dryad.v6wwpzh6p.

**Funding:** IDW, MS, MSF - IDW (1), MS (1), MSF (1) - Project Apis m. - https://www.projectapism.org/ - The sponsor did not play a role in the study design, data collection and analysis, decision to publish or preparation of the manuscript.

**Competing interests:** The authors have declared that no competing interests exist.

## Introduction

Social insects such as honey bees (*Apis mellifera)* have heritable behavioral defenses to resist pathogens and parasites [1]. This ability to behaviorally reduce the transmission of pathogens and parasites within their densely packed populations is called social immunity [1]. One form of social immunity is hygienic behavior [2], which was first described in honey bees as the ability to detect, uncap and remove brood (larvae and pupae) infected with American foulbrood (caused by the bacterium *Paenibacillus larvae*) from the nest [2,3]. Later studies found a correspondence between honey bee hygienic behavior and resistance to chalkbrood disease (caused by fungus *Ascosphaera apis*) [4–6]. Bee brood is physiologically susceptible to both pathogens, but the hygienic removal occurs before the pathogen reaches the infectious stage. At this stage, the larva or pupae may show early signs of disease that are not easily diagnosed in the field. The rapid detection and hygienic removal of the brood by the bees eliminates or reduces disease transmission [2,3,7]. Hygienic behavior is also one form of resistance to the parasitic mite, *Varroa destructor* as the bees detect and remove the pupae on which the mite reproduces [8,9]. Removal of mite-infested pupae interrupts the reproductive cycle of the foundress mite [10], which eventually can lead to foundress reduced fertility [11,12].

Hygienic behavior can be selectively bred to form distinct lines of honey bees that demonstrate natural resistance to pathogens and parasites [2,4,9,13,14]. One example was the MN Hygienic line bred at the University of Minnesota from 1994–2009 [9,13]. This line was selected based on the ability of colonies to rapidly remove freeze-killed brood (FKB) from the nest; the dead brood serves as a proxy for diseased brood [13]. After field challenge with a pathogen, colonies in the MN Hygienic line showed resistance to American foulbrood (AFB) and chalkbrood [13]. The MN Hygienic line also detected and removed *Varroa* mites after challenge [9]. Field trials showed that the mite levels in MN Hygienic colonies were significantly lower compared to unselected colonies of commercial origin. However, the MN Hygienic line was not considered fully resistant to mites as it required treatment to keep mite levels below a treatment threshold [15,16].

Beginning in the 1990s, another line of bees was bred at the USDA-ARS Bee Research Lab, specifically for resistance to *Varroa* [11,12,16,17]. The FKB assay was not used to select for resistance to *Varroa*; instead, mite resistance was selected more directly by quantifying the ability of colonies to maintain low levels of mites over time using various metrics; e.g., mite population growth over the season, removal rate of mite-infested pupae, and mite fertility (reviewed in [17]). To ensure the line was commercially viable, the USDA-ARS outcrossed VSH-queens in some bee operations, and then selected colonies based on high honey production, large populations for pollination, and mite resistance. In 2011, the selected population was renamed "Pol-line Hygienic Italian honey bees" (Pol-line) [18]. The Pol-line stock demonstrates significant expression of mite resistance and is still maintained by the USDA-ARS [19].

The terms and acronyms for hygienic traits and lines of honey bee merits clarification. The acronym VSH is used both for the line of bees bred by the USDA-ARS

lab, and to describe the trait, *Varroa* Sensitive Hygiene, in which honey bees specifically detect and remove mite-infested brood [10,20,21]. The term "general hygiene" was introduced [22] to distinguish it from VSH. Colonies selected for general hygiene are considered to demonstrate disease resistance but not mite resistance [16,17]. While the behavioral patterns for VSH and general hygiene are the same (detection, uncapping and removal), it is not clear if their neural and genetic underpinnings are similar [23–25]. Though they do share some overlapping genetic regions, suggesting some shared genetic characteristics [14].

One approach to exploring potential differences between VSH and general hygienic behavior is to test their behavioral responses to both pathogens and mites in the field. While FKB results generally correlate with overall disease resistance, it is not always accurately predictive of a colony's ability to fight brood pathogens, thus field testing with actual disease agents is particularly important for holistic assessment [26,27]. It has not been tested if the Pol-line (with VSH trait) is specifically resistant to mites; that is, whether the hygienic behavior of the VSH trait is sensitive only to *Varroa*, or whether their hygienic behavior extends to other brood diseases. The goal of this study was to explore the relationship between honey bee resistance to the parasitic mite *Varroa* and resistance to the pathogen *Ascosphaera apis* that causes chalk-brood disease. We challenged colonies selected for VSH (Pol-line) and colonies selected for general hygiene by the FKB assay (Commercial line) with spores of *A. apis* and compared signs of chalkbrood disease over three trials and in two locations (Minnesota and Baton Rouge). We also explored the correspondence between each line's hygienic response to the FKB assay and their mite loads. We hypothesized that the Pol-line, although bred specifically for mite resistance, would demonstrate resistance to both chalkbrood and mites, while the Commercial line would demonstrate chalkbrood resistance but not mite resistance.

## Methods and materials

### Study design and timeline overview

This study was conducted over two years in Minnesota (2023 and 2024) and was replicated in Baton Rouge (2024). Two queen types were used: (1) Commercial queens from Northern California bred for hygienic-behavior using the freeze-killed brood (FKB) assay and (2) Pol-line queens bred for VSH from the USDA-Bee Breeding Lab in Baton Rouge using *Varroa*-specific assays [18]. Detailed methods are described below; differences in methods between years and locations are summarized in S1 Table.

### Queen sources and packages

In April 2023, we purchased packages of bees (containing about 10,000 adult worker bees and a caged queen) from a commercial queen producer in northern California. The producer selected for hygienic behavior using the freeze-killed brood assay (FKB) as described in [28] and briefly below. Half of the caged queens that came with the packages were removed and replaced with queens from the Pol-line obtained from the USDA-ARS Bee Breeding laboratory in Baton Rouge. The queens in the remaining packages were left in place. All packages were hived in standard Langstroth deep boxes containing new foundation to ensure no disease spores were present in the wax combs. After hiving, the caged queens were released into the hives. All colonies were supplemented with a pollen patty and sugar syrup for three weeks to ensure colony growth. No colonies were treated for mites in the spring.

In 2024, packages were made at the University of Minnesota by first shaking bees from healthy, wintered colonies into a large screen shaker box to homogenize the source of bees, and then placing 1.13 kg of bees into individual package boxes. Each package received either a Pol-line queen from the USDA-ARS Bee Breeding Lab Laboratory in Baton Rouge or a commercial queen from the same queen producer from northern California that was used in 2023. Each package was hived in new Langstroth boxes with frames containing new foundation and later the caged queens were released. All colonies were supplemented with sugar syrup and pollen patties for approximately three weeks to support colony growth.

Seven days after the colonies were established, they were treated with 50 ml oxalic acid in sugar water to equalize mite loads among colonies as 17% had over two mites/100 bees.

In 2024 at the USDA-ARS Bee Breeding Lab in Baton Rouge Lab, colonies were established by dividing healthy colonies and requeening each new colony with either Pol-line queens from the Baton Rouge Lab or commercial queens from the same queen producer in northern California as used in Minnesota. No colonies were treated for mites in the spring. The Pol-line queens used in Baton Rouge were reared and naturally mated during the last week of March 2024 in an area near Baton Rouge saturated with drones from other Pol-line queens. The Pol-line queens sent to the University of Minnesota were also reared in the same location in Baton Rouge but mated during mid-April 2023 and 2024.

In 2023 and 2024, the colonies were located in Carver Park Reserve, Victoria, Minnesota. Colonies were set up in circles with entrances facing out and in full sun as much as possible. Queen line was alternated among colonies. In Baton Rouge, colonies were set up on pallets at the USDA-ARS Bee Breeding Lab, alternating by queen line and direction. In 2023, all colonies were maintained in two deep boxes. In 2024 in both locations, all colonies were maintained in one deep box and provided additional boxes over a queen excluder as needed. Sample sizes are given in Table 1.

## Chalkbrood challenge

In 2023, 230 fresh chalkbrood mummies (i.e. pre-pupae infected with *A. apis* spores) were homogenized and mixed into mixed-source pollen and 50% sucrose solution (following [4,29]). Six weeks after the queens were introduced into the colonies, all colonies were given a 0.23 kg pollen patty containing 3.3 mummy equivalents. Each mummy can contain millions of spores [30] but the amount fed to larvae by the adult bees could not be determined.

To increase infection likelihood in 2024, we challenged the larvae directly using a spore spray solution in both locations, six weeks after queens were introduced. *A. apis* spores were homogenized and mixed into PBS buffer, adapting methods from [4]. Two to three frames of open brood were sprayed with a total of 100 mL of the spore spray solution that had a concentration of $1.5 \times 10^6$ *A. apis* spores/mL seven weeks after queens were introduced. Spores were propagated

**Table 1. Sample sizes (number of colonies) and prevalences (presence/absence) of chalkbrood mummies and early signs of chalkbrood.**

| Location & Year | Days post-challenge | Sample size | | Prevalence mummies | | Prevalence early signs | |
|---|---|---|---|---|---|---|---|
| | | Pol | Comm | Pol | Comm | Pol | Comm |
| Minnesota 2023 | 0 d | 11 | 12 | – | – | – | – |
| | 7 d | 10 | 12 | 5 | 6 | – | – |
| | 14 d | 8 | 12 | 3 | 3 | – | – |
| Minnesota 2024 | 0 d | 14 | 10 | – | – | – | – |
| | 2 d | 14 | 9 | 0 | 0 | 4 | 2 |
| | 4 d | 9 | 8 | 5 | 5 | 7 | 8 |
| | 7 d | 9 | 8 | 6 | 5 | 8 | 7 |
| | 14 d | 8 | 7 | 3 | 3 | 6 | 3 |
| Baton Rouge 2024 | 0 d | 13 | 14 | – | – | – | – |
| | 2 d | 13 | 14 | 1 | 0 | 12 | 11 |
| | 4 d | 13 | 14 | 2 | 8 | 11 | 14 |
| | 7 d | 13 | 14 | 0 | 1 | 8 | 6 |

Day zero (0 d) is the day of challenge. Prevalence was compared between queen types by binomial regression at each time point. In Minnesota, sample sizes decreased due to queen issues (e.g., supersedure, queen loss). Pol = Pol-line, the mite resistant line; Comm = Commercial line bred for hygienic behavior.

on media according to standard methodology (described in [30]) chalkbrood mummies collected in North Dakota in 2023. Spores were harvested after they reached the black growth stage and then used at both locations in 2024.

## Quantification of chalkbrood challenge in colonies

**Chalkbrood challenge.** In 2023, colonies were inspected seven and 14 days post-challenge. To ensure that early signs of infection were not missed, in 2024 colonies in Minnesota were inspected two, four, seven, and 14 days post-challenge. Colonies in Baton Rouge were inspected at two, four, and seven days post-challenge, after all signs of infection had cleared. At each inspection, each brood frame (front and back) was given a ranked score to quantify the number of chalkbrood mummies observed. A score of 0 indicated that no cells containing chalkbrood mummies were observed per comb; 1 = 1–5 mummies per comb; 2 = 6–25 mummies per comb; and 3 ≥ 25 mummies per comb [13]. On each inspection date, an overall severity score for each colony was obtained by calculating the mean (± s.d) of the individual comb scores. An overall score of 1 corresponded to a colony with only slight clinical signs, possibly not noted by cursory inspection. An overall score of 2 indicated noticeable signs of infection, and a score of 3 corresponded to a highly infected colony [13].

**Early signs.** Early signs of chalkbrood were included in the metrics for measuring chalkbrood infection in 2024 in both locations. Early signs of chalkbrood included acuminate and amorphous pre-pupae in which the cell's wax capping has been removed by the bees (Fig 1). Pre-pupae under cells that had evidence of uncapping [31] were collected for RT-qPCR confirmation of chalkbrood, in addition to acuminate and amorphous pre-pupae (Fig 1).

To confirm that the malformed pre-pupae had chalkbrood and not sacbrood virus, we performed a RT-qPCR analysis. Pooled bee samples representing a single colony and early chalkbrood sign (S2 Table for sample size) were transferred to a 50 ml sterile gentle MACS™ M Tube, 15 ml of sterile 1x PBS was added and the samples were dissociated with gentleMACS Dissociator (V1.02, Miltenyi Biotec Inc. Auburn, CA, USA, RNA_02.01). Samples were further prepped for viral RNA extraction following protocol by Nikulin et al. 2024 [32]. Viral RNA extraction was done via automated extraction with NucleoMag Virus RNA-DNA Isolation kit by Takara and Magnetic Particle Processor (MMP) (KingFisher Flex, ThermoScientific) following manufacture and user manual instructions. RNA purity and concentration was assessed via NanoDrop Sepctrophotometer (NanoDrop™, Thermo Fisher Scientific, USA). To screen for viral loads, Power-UP SYBER Green RNA-to-CT 1 Step it (Applied biosystems, Foster City, CA, USA) was used. To quantity viral loads in the RNA samples 5

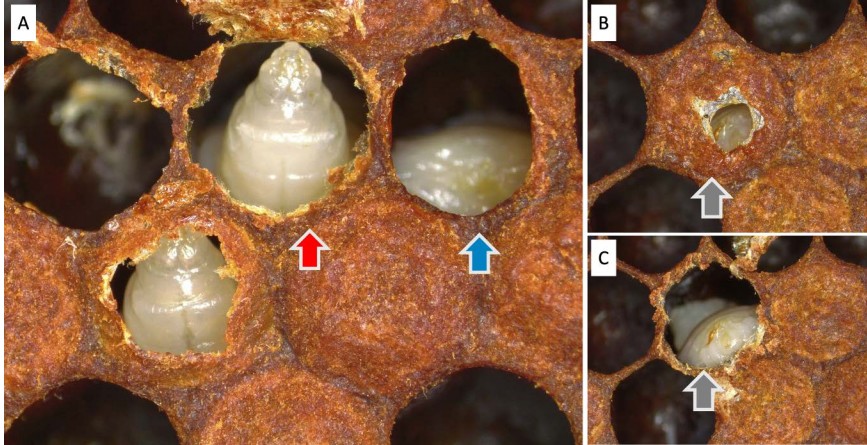

**Fig 1. Early signs of chalkbrood.** (A) Acuminate heads (red arrow) and amorphous pre-pupae (blue arrow) are both are early signs of non-infectious chalkbrood. Grey arrows show a cell where worker bees have started to remove the wax capping of cells (B), revealing a pre-pupae underneath that had early signs of chalkbrood (C).

ul of SYBER mix, 3.92 ul of RNA diluted in molecular grade water to 50 ng/ul, 0.05 ul reverse primer of selected virus, and 0.05 ul of forward primer of selected virus were used [32]. The proper primers and RT-qPCR denaturing and annealing temperature protocols were used for chalkbrood and sacbrood virus (S3 Table). Viral genome copies were calculated [33] and expressed as log10 viral RNA copies per sample.

**Peak infection.** The scores for early signs and mummies were summed to give a peak severity score at one week post-challenge in 2024. In 2023, only mummies were summed at one week as early signs were not recorded.

**Colony assessments.** Prior to chalkbrood challenge, colony strength was recorded by counting the number of frames completely covered with bees as well as the number of frames with brood. Brood pattern was rated on a scale of 1–5 with 1 being poor and 5 being excellent. Any signs of brood disease (i.e., unusual color or form) were noted. Colonies were equalized on the day they were hived in terms of bees and brood in Minnesota and Baton Rouge prior to inoculation.

**Freeze-killed brood assay.** Six weeks after the experimental queens were laying in the colonies and all brood and adult bees were progeny of the new queens, the level of hygienic behavior in each colony was assayed using the freeze-killed brood (FKB) test. Following standard methods outlined in [28], liquid nitrogen was poured into a 7.62 cm PVC tube to freeze-kill 160 sealed pupae in the purple-eye stage of development. After 24 hours, the number of pupae that were completely removed and were in the process of being removed (partially removed) by each colony were quantified. The "strict" test for hygienic behavior records if a colony completely removes > 95% of the FKB within 24 hours [17]. The "liberal" test includes the number of pupae that are both fully and partially removed within 24 hours.

***Varroa* mites.** Samples of adult bees were collected from all colonies pre-chalkbrood challenge and at the end of the season (June and early September) to estimate the amount of *Varroa* mites on adult bees in each colony. In Baton Rouge, mite samples were collected pre-challenge in June. A sample of 300 bees from a comb containing all stages of brood (eggs, larvae, and pupae) were collected in a 20 ml screw cap vial containing 70% ethanol [34]. In the lab, the bees and mites in the vials were processed to extract mites with an alcohol wash method [35]. The mites and bees were separated and counted by placing each sample in a shallow dish filled with 70% ethanol, shaking it for 60 seconds, and straining it through coarse hardwire cloth repeatedly until no mites were found in the dish on two consecutive washes. In each colony, the average weight of each individual bee was estimated by weighing 100 bees per sample and dividing the weight by 100. The whole weight of the sample was obtained thereafter. To calculate the total number of bees in the sample, the weight of the entire sample was divided by the average weight per bee. Mite infestation was calculated by taking the total number of mites counted over the total bees in the sample [34].

## Statistics

Data were analyzed using R statistical software, version 4.3.1. We used the *performance* package [36] to check model assumptions and *ggpplot2* [37] to visualize results. For every analysis, data were analyzed separately by location and year unless specified. Detailed analyses with associated packages are as follows.

As the data were non-normal, a Kruskal-Wallis test was used to assess if there were any differences in chalkbrood loads detected by real-time RT-qPCR between the early sign categories: acuminate, amorphous, and uncapped pre-pupae.

The prevalence of chalkbrood mummies and early signs were compared between queen types (Pol-line or Commercial) using binary logistic regression with chalkbrood presence or absence as the response variable and queen line as the predictor variable. A Firth Penalized Regression model was used if there was perfect separation of the variables. All other variables (severity scores for chalkbrood mummies and early signs, hygienic behavior, and mite loads) were compared between queen types by analysis of variance (ANOVA). If the data were non-normal and did not meet heteroscedasticity requirements, a Kruskal Wallis test was used in place of ANOVA. A t-test was used to compare pre challenge and post challenge mite loads. If the data were non-normal, a Wilcoxon test was used instead.

Differences in frames of brood or frames of bees between queen types for 2023 and 2024 was analyzed using a Wilcoxon test as the data were non-parametric. To investigate how chalkbrood severity differed between 2023 and 2024 in Minnesota at 7 days and 14 days, a Kruskal-Wallis test was used as the data was non-normal and did not meet heteroscedasticity requirements.

To investigate how chalkbrood mummy and chalkbrood early sign severity were affected by hygienic behavior, mite infestation, and queen line, we used general linear models. If the data were non-normal a Kruskal Wallis test was used in place of ANOVA or a Spearman correlation was used in place of linear models.

## Results

### Chalkbrood infection

**Prevalence.** In 2023, half of the colonies developed signs of chalkbrood mummies at seven days post challenge in Minnesota. Mummy prevalence between Pol-line and Commercial colonies at seven days or 14 days was not significantly different post challenge (Table 2, S4 Table) The prevalence of early signs of chalkbrood was not recorded in 2023.

Minnesota colonies in 2024 showed no significant difference in prevalence of mummies or early signs between the Pol-line and Commercial colonies at any inspection date post-challenge. In Baton Rouge, there was no significant difference in prevalence of mummies or early signs at two and seven days post challenge. At four days there was no difference in prevalence of early signs between the two queen types. However, Pol-line colonies had significantly (86.4%) lower odds of having chalkbrood mummies compared to Commercial colonies ($\beta = -1.99$, $p = 0.034$) (Table 2, S4 Table).

**Chalkbrood mummy severity.** Throughout both years in Minnesota, no chalkbrood was detected by RT-qPCR prior to challenge. In 2023 there was no significant difference in chalkbrood severity or queen line at seven days or 14 days post challenge (Fig 2, Table 2). In 2024 there was no significant difference in severity between the Pol-line and Commercial line at any time point in Minnesota (Fig 3, Table 2). However, at four days post challenge in Baton Rouge, the Pol-line had significantly lower chalkbrood severity compared to the Commercial colonies ($\chi^2(1) = 4.5$, $p = 0.033$) (Fig 4). This relationship was not seen at two days or seven days post challenge (Fig 4, Table 2).

**Early signs of chalkbrood severity.** Early signs of chalkbrood were confirmed with RT-qPCR for amorphous and acuminate pre-pupae. Pre-pupae under cells that were disturbed (i.e., uncapped) were also positive for chalkbrood (Fig 5). No sacbrood was detected. There was no significant difference between the chalkbrood titers and category of early sign (acuminate, amorphous, recapped) ($\chi^2(2) = 0.85$, $p = 0.65$). RT-qPCR showed that sacbrood was not detected in the any of the pre-pupae.

**Table 2. Chalkbrood and early sign severity by queen line.**

| Location & Year | Days post challenge | Severity of chalkbrood mummies by queen | | | | Severity of early signs by queen | | | |
|---|---|---|---|---|---|---|---|---|---|
| | | $\chi^2$ | F | df | p | $\chi^2$ | F | df | p |
| Minnesota 2023 | 7 d | 0.32 | – | 1 | 0.57 | – | – | – | – |
| | 14 d | 0.21 | – | 1 | 0.64 | – | – | – | – |
| Minnesota 2024 | 2 d | – | – | | | 0.04 | | 1 | 0.83 |
| | 4 d | – | 0.72 | 1,15 | 0.41 | – | 1.58 | 1,15 | 0.22 |
| | 7 d | 0.06 | – | 1 | 0.81 | – | 1.70 | 1,15 | 0.21 |
| | 14 d | 1.41 | – | 1 | 0.23 | 0.7 | | 1 | 0.40 |
| Baton Rouge 2024 | 2 d | 0.002 | – | 1 | 0.95 | – | 0.96 | 1,25 | 0.33 |
| | 4 d | 4.5 | – | 1 | 0.03* | – | 10.8 | 1,15 | 0.003** |

The only significant difference was found at four days post challenge in Baton Rouge where the Pol-line had significantly lower early sign severity compared to the Commercial line (F(1,15) = 10.8, p = 0.003, $\eta^2 = 0.30$). d = days.

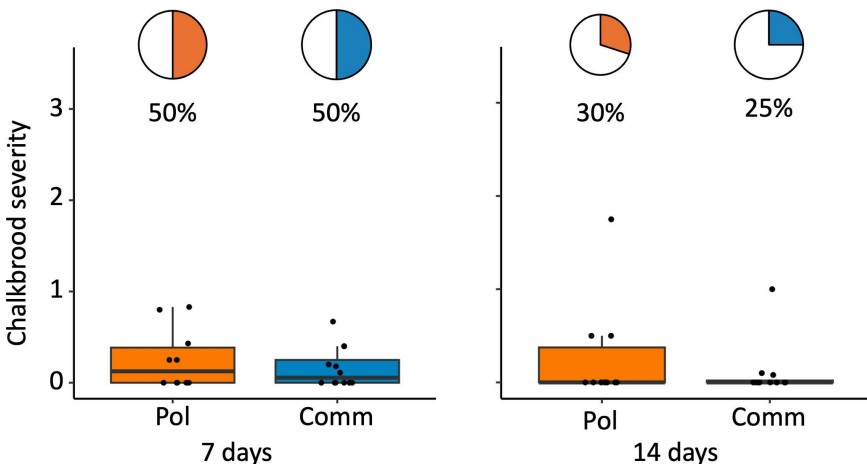

**Fig 2. 2023 Chalkbrood mummy prevalence and severity.** Pie charts show percentage of colonies with chalkbrood mummies (prevalence), and respective box plots the severity of chalkbrood. There was no difference in chalkbrood mummy prevalence or severity between the two queen types (Table 2 and S4 Table for stats). Pol = Pol-line, the mite resistant line; Comm = Commercial line bred for hygienic behavior.

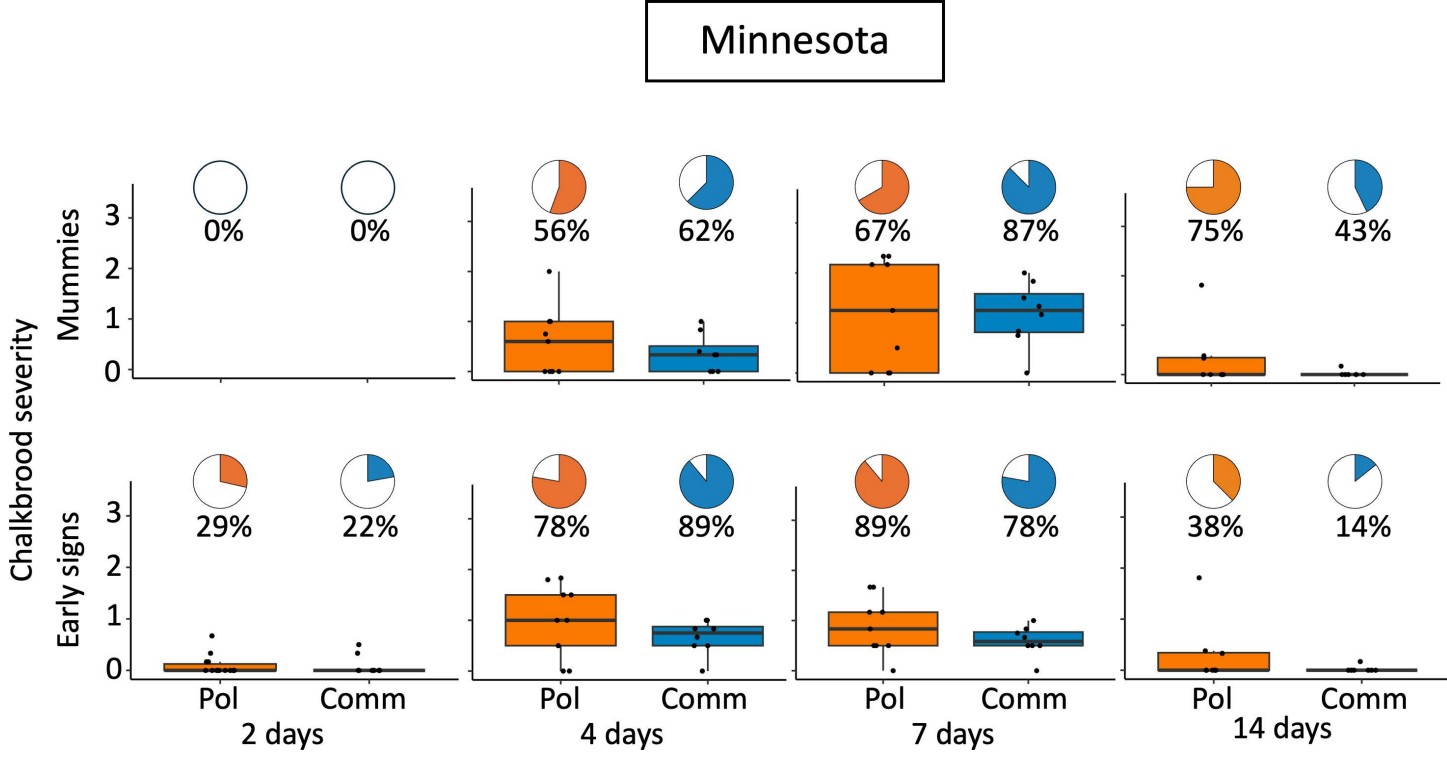

**Fig 3. 2024 Minnesota chalkbrood early signs and mummy severity.** Pie charts show the percentage of colonies with chalkbrood mummies or early signs (prevalence), and respective boxplots represent the severity of chalkbrood mummies (top row) or early signs (bottom row). There was no statistical difference in the prevalence or severity of chalkbrood mummies or early signs between queen line (Table 2 and S4 Table for statistics). Pol = Pol-line, the mite resistant line; Comm = Commercial line bred for hygienic behavior.

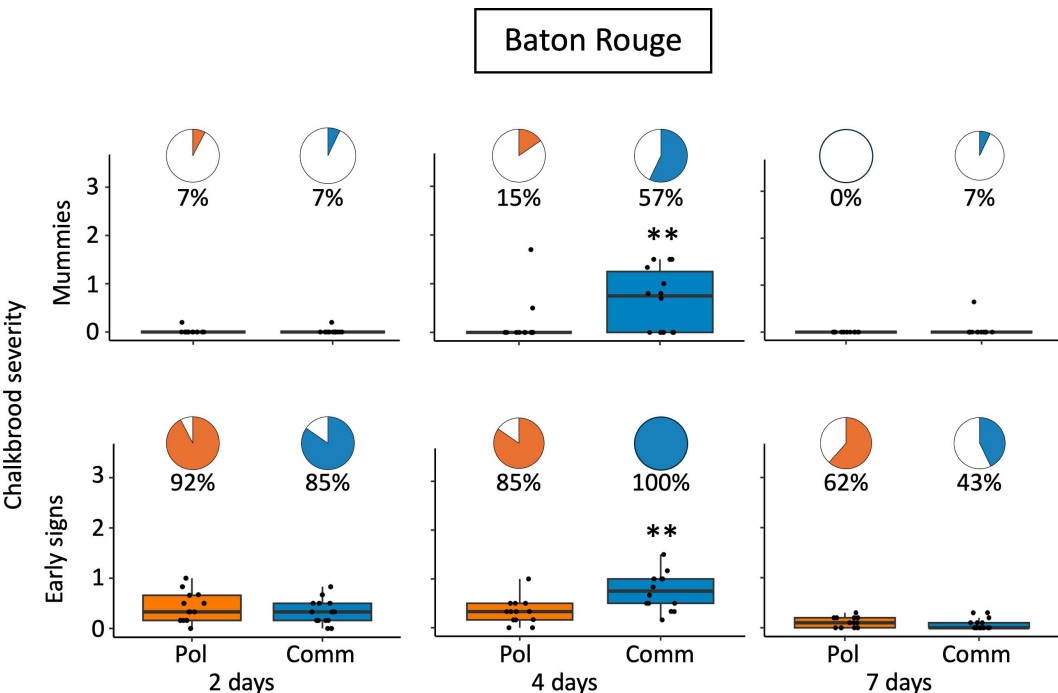

**Fig 4. 2024 Baton Rouge chalkbrood early signs and mummy severity.** Pie charts show the percentage of colonies with chalkbrood mummies or early signs (prevalence), and respective boxplots represent the severity of mummies (top row) or early signs (bottom row). The only significant difference between queen lines and chalkbrood mummies was at four days where the Pol-line had significantly lower mummy severity (p = 0.033, ε² = 0.14) and 86.4% lower odds of having chalkbrood mummies. The Pol-line also had significantly lower early sign severity (p = 0.003, η² = 0.30) compared to the Commercial line at four days post challenge (Table 2 and S4 Table for statistics). Pol = Pol-line, the mite resistant line; Comm = Commercial line bred for hygienic behavior.

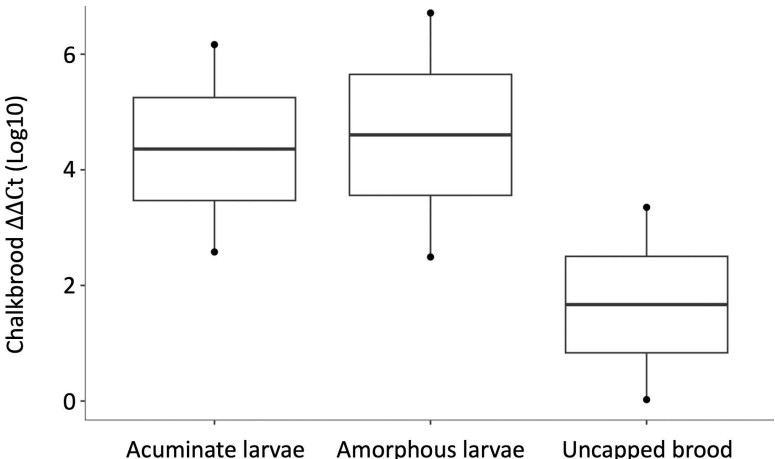

**Fig 5. Detection of chalkbrood in acuminate and amorphous pre-pupae and pre-pupae under uncapped cells.** Uncapped cells are cells where worker bees have started to remove the wax capping of cells. One dot represents 12-20 pooled pre-pupae exhibiting early signs of chalkbrood (S2 Table for number of pre-pupae pooled).

There were no differences in early sign severity scores and queen lines in Minnesota for any quantification date (Fig 3, Table 2). The only significant difference in Baton Rouge was found at four days post challenge the Pol-line had significantly lower early sign severity compared to the Commercial line (p = 0.003, η² = 0.30) (Fig 4).

**Peak infection, challenge method, and colony strength.** In Minnesota, the highest severity scores (peak infection) for chalkbrood were at seven days post-challenge in both 2023 and 2024 (Figs 2 and 3). In Baton Rouge, the peak infection were observed four days post-challenge (Fig 4). At day seven, Minnesota had significantly higher chalkbrood mummy severity in 2024 compared to 2023 (χ²(1) = 12.84, p = 0.0003) (Fig 6). By 14 days, there was no difference in chalkbrood severity between the two years in Minnesota as nearly all colonies had recovered by then (χ²(1) = 0.046, p = 0.83). In 2023, the Pol-line colonies had significantly fewer frames of bees on the day of challenge; however, there was no difference in frames of brood or brood pattern. In 2024, there were no differences in frame of bees, frames of brood, or brood pattern between the Pol-line and Commercial colonies (S5 Table for statistics).

## Hygienic behavior, mites, and FKB

**Hygienic behavior.** In Minnesota 2023, there was no difference between queen lines and the percent of FKB removed in 24 hours by either strict or liberal testing (Table 3) (Fig 7A). In 2023, 16% Commercial colonies and 18% of the Pol-line colonies completely removed > 95% FKB by the strict test. However, 50% of Commercial colonies and 70% of Pol-line colonies in 2023 scored > 95% of the FKB based on the liberal test

In 2024, the Pol-line removed significantly less FKB compared to the Commercial line on both strict (χ²(1) = 8.54, p = 0.003) (Fig 7A) and liberal tests (F(1,15) = 5.33, p = 0.03, η² = 0.26). Over half (55% colonies) of the Commercial line removed over 95% of the FKB within 24 hours. In contrast, no colonies within the Pol-line removed > 95% of the FKB within 24 hours (Pol-line mean score ± sd = 63% ± 16, Comm 83% ± 25). In Baton Rouge there was no difference between Pol-line and Commercial line by the strict or liberal FKB tests with 60% of each queen line scoring > 95% strict (Table 3 for stats) (Fig 7A).

**Mites.** In 2023, *Varroa* loads (mites/100 adult bees) increased between June and September in both Pol-line and Commercial colonies (Pol-line p = 0.008, Comm p = 0.0003) (Table 3, Fig 7B). In June, Pol-line had significantly fewer mites compared to Commercial colonies (p = 0.005). This trend persisted in September (p = 0.003) (Table 3, Fig 7B).

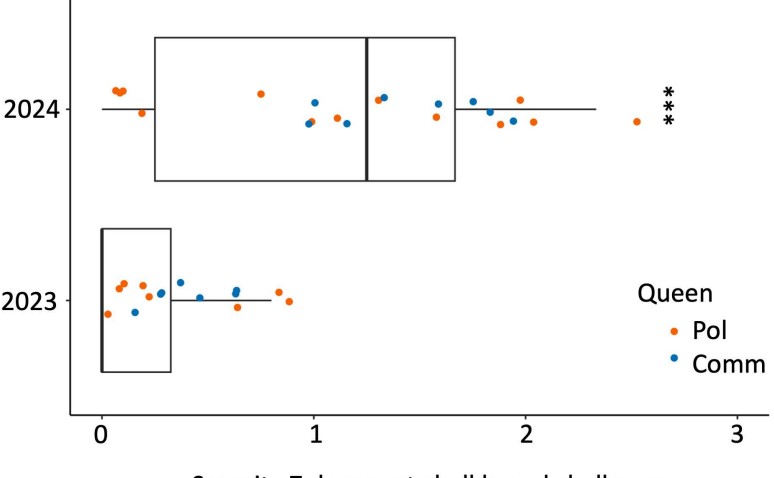

**Fig 6. Higher chalkbrood mummy severity in 2024 vs. 2023.** There was significantly higher chalkbrood mummy severity in Minnesota 2024 at seven days post challenge compared to Minnesota 2023 (p = 0.0003). Pol = Pol-line, the mite resistant line. Comm = Commercial line bred for hygienic behavior.

**Table 3. FKB, mite infestation, and peak chalkbrood severity.**

| Location and Year | FKB by queen | | | Difference in mite infestation by queen | | | | | | Difference in mite infestation June to September | | | | Peak chalkbrood severity by queen | | |
|---|---|---|---|---|---|---|---|---|---|---|---|---|---|---|---|---|
| | | | | June | | | September | | | | | | | | | |
| | $x^2$ | df | p | $x^2$ | df | p | $x^2$ | df | p | Queen | t | df | p | $x^2$ | df | p |
| Minnesota 2023 | 0.12 | 1 | 0.74 | 7.72 | 1 | 0.005** | 12.90 | 1 | 0.0003*** | Pol | 3.46 | 1 | 0.008** | 4.08 | 2 | 0.13 |
| | | | | | | | | | | Comm | 5.12 | 1 | 0.0003*** | | | |
| Minnesota 2024 | 8.53 | 1 | 0.003** | 0.18 | 1 | 0.67 | 0.018 | 1 | 0.89 | Pol | −2.0 | 6 | 0.08 | 0.0005 | 1 | 0.98 |
| | | | | | | | | | | Comm | −0.95 | 6 | 0.37 | | | |
| Baton Rouge | 0.03 | 1 | 0.86 | 4.92 | 1 | 0.02* | — | — | — | Pol | — | — | — | 10.7 | 1 | 0.001** |
| | | | | | | | | | | Comm | — | — | — | | | |

In 2024 Minnesota, the Pol-line had significantly lower FKB scores compared to the Commercial line (p = 0.003). Pol mite infestation levels were significantly lower in Minnesota 2023 June (p = 0.005) and September (p = 0.0003) and in Baton Rouge in June (p = 0.02). By September 2023, mite levels had significantly increased for both the Pol-line (p = 0.008) and the Commercial line (p = 0.0003). Pol-line had the same amount of chalkbrood (p = 0.13, p = 0.98) or significantly less chalkbrood (Baton Rouge, p = 0.001) compared to the Commercial colonies. Comm = Commercial line bred for hygienic behavior; Pol = Pol-line, the mite resistant line; FKB = Freeze-killed brood.

In 2024, the Pol-line and Commercial colonies in Minnesota showed no significant differences in mite infestation in June or September (Fig 7B). Additionally, there was no significant increase in mite infestation from June to September for either queen line (Table 3). In Baton Rouge, the Pol-line had significantly lower mites/100 adult bees compared to the Commercial colonies (p = 0.026) (Table 3, Fig 7B). Mite samples were not taken in the fall in Baton Rouge.

**Freeze-killed brood as a predictor of chalkbrood mummies and mites.** The level of hygienic behavior, as measured by the FKB assay, was not a significant predictor of chalkbrood prevalence or severity in 2023 and 2024 in Minnesota and Baton Rouge (S6 Table). Additionally, FKB was not a significant predictor of peak chalkbrood severity (mummies + early signs) in Baton Rouge (ρ = −0.13, p = 0.49) and Minnesota 2024 (ρ = 0.12, p = 0.62), or of mite infestation throughout either year in both locations (Fig 7, S7 Table).

## Discussion

The aim of this study was to explore the behavioral differences in mite resistance and disease resistance between colonies bred for *Varroa* Sensitive Hygiene (VSH) and colonies that showed general hygiene. Supporting our hypothesis, we found that colonies from the Pol-line bred for VSH were as resistant to chalkbrood as bees from a commercial line bred for hygienic behavior by the FKB assay. Additionally, the Pol-line was as hygienic as the Commercial line and had significantly lower mite loads in two of three trials. These results indicate that honey bees selected for the VSH-trait respond to both mite-infested and disease-infected brood.

The neural and genetic mechanisms underlying the VSH-trait and general hygienic behavior have not been fully resolved [17,20,38]. Hygienic bees selected using the FKB assay have higher olfactory sensitivity to diseased brood compared to non-hygienic bees [25,39–42]. Bees selected with FKB also have unique protein expression involved in the cascade of events required to respond to olfactory signals from diseased or dead brood [43]. Bees selected for VSH also have a high degree of olfactory sensitivity and are more sensitive to changes in brood odors than bees without the VSH trait [25,44,45]. The genetic bases of general hygienic behavior and VSH are both considered quantitative [23,46,47]; c.f., [2]). However a meta-analysis of genomic, transcriptomic, and proteomic (-omic) studies revealed a lack of uniformity in the genes controlling these traits [38]. Despite the lack of congruence, functions associated with some genetic markers demonstrated consistency between VSH and general hygienic behavior (e.g., neural sensitivity, signal transmission, sensory perception, and olfaction; [14,38]). Future research is needed to determine if bees selected for VSH are sensitive and responsive to odors of any brood that is abnormal, whether freeze-killed, diseased or mite-infested. This could help inform and interpret -omic studies.

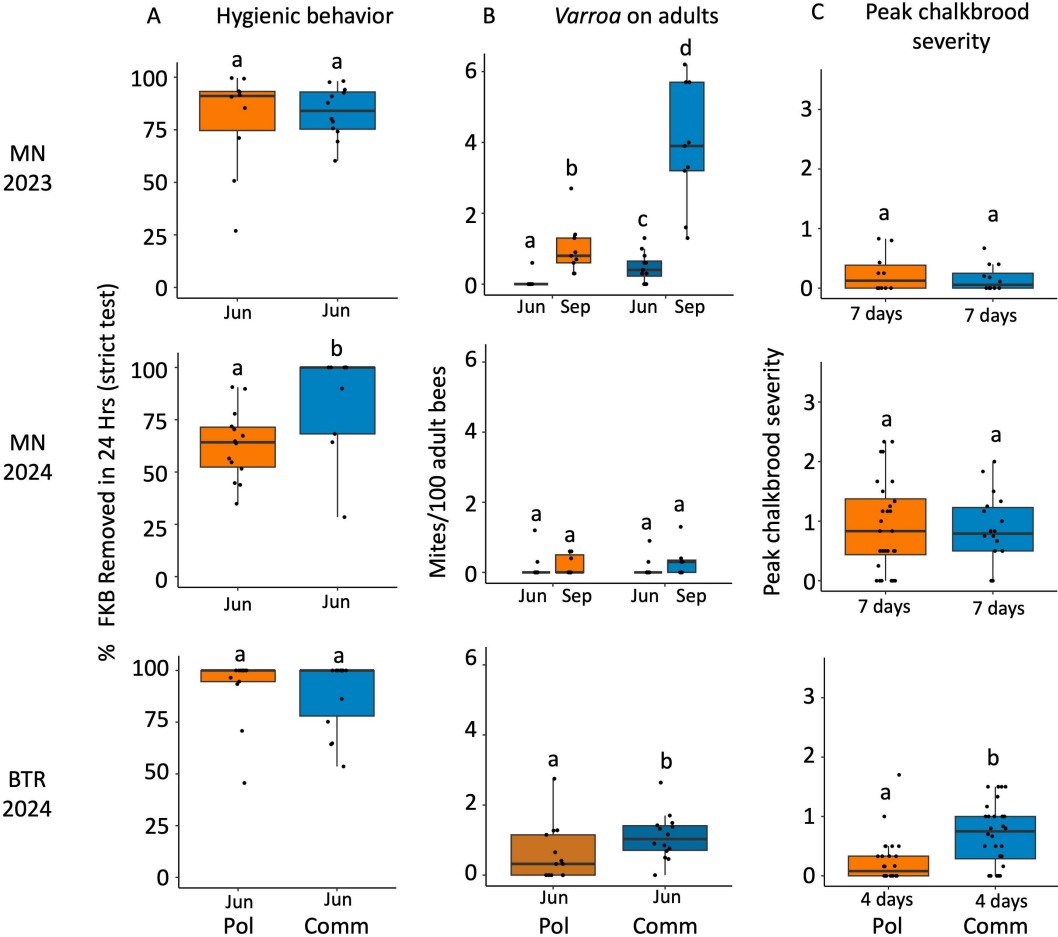

**Fig 7. Comparison of hygienic behavior based on strict % FKB (column A), Varroa on adults based on mite counts per 100 adult bees (column B), and peak chalkbrood severity (column C) across trials. Hygienic behavior**: In MN 2024, the Pol-line was significantly less hygienic (p = 0.003) compared to the Commercial line, but there were no significant differences between the queen lines in MN 2023 and BTR 2024 (A). ***Varroa* on adults:** The Pol-line had significantly fewer mites compared to the Commercial line in MN 2023 in June (p = 0.005) and September (p = 0.0003), and in BTR 2024 in June (p = 0.02) but not in MN 2024 Mite loads were higher for both queen types in Sep compared to Jun in MN 2023 (Pol-line: p = 0.008, Commercial: p = 0.0003) but not in MN 2024 when there was no significant difference in mites between queen lines (B). **Peak chalkbrood severity:** The Pol-line had significantly lower chalkbrood severity at peak infection in BTR in 2024 (p = 0.001), but there were no differences in severity in MN either year (C). See Table 3 for statistics. FKB = freeze-killed brood; Jun = June; Sep = September; BTR = Baton Rouge; MN = Minnesota; Comm = Commercial line bred for hygienic behavior; Pol = Pol-line, the mite resistant line.

## Chalkbrood disease

In nature, chalkbrood spores can be transmitted to the brood by the adult bees, the carriers of the pathogen, when they feed larvae (40,41). Experimentally, honey bee colonies can be challenged with chalkbrood spores by feeding adult bees pollen containing spores, or by direct application of spores onto larvae within the comb [48]. In 2023, we inoculated colonies with pollen "patties" containing approximately 3 million spores but the dose the larvae received by the adult bees through food transfer was unknown. In 2024, each colony was given 100 million spores sprayed directly onto first to third instar larvae. The results showed that the prevalence and severity of chalkbrood mummies were higher in 2024 compared to 2023 (Fig 6), indicating that the direct challenge was more effective in inducing infection at least with the *A. apis* strains used in this study.

This study is the first documentation of the early signs of chalkbrood. The presence of chalkbrood mummies in comb indicates that the pathogen has reached the infectious, spore stage. In 2023, we observed pre-pupae with early signs of disease, appearing as uncapped pre-pupae with pointed heads (acuminate) and/or misshapen (amorphous) (Fig 1). These early signs appear similar to early stages of sacbrood virus [49,50], but can be distinguished using rt-qPCR (Fig 5). In time, sacbrood infected brood becomes discolored within a fluid-filled sac, whereas chalkbrood infected brood develops visible signs of fungus (becomes mummified) beginning on the posterior end of the brood [51]. The early signs we observed were likely the non-infectious stages of chalkbrood, which hygienic bees detect and remove from the colony preventing disease transmission [3,7]. Future studies could monitor the bees' removal of non-infectious stages of chalkbrood to more accurately assess behavioral resistance through hygienic behavior.

In addition to hygienic behavior, honey bees may have varying degrees of physiological resistance to *A. apis* [52]. Some studies have identified potential genomic markers associated with larval resistance to chalkbrood but the studies lack consistency as to where markers involved in resistance are located [53–55]. Follow-up studies could be done to verify genomic markers and explore the combined roles of physiological resistance and hygienic removal of early stages of infection.

## Mites

Past research has found that the Pol-line is highly mite resistant [19]. Our study confirms these findings as the Pol-line had significantly lower mites compared to the Commercial line in two of the three trials. It remains unclear if this difference in *Varroa* resistance is driven by distinct different genetic mechanisms between VSH and general hygiene, or if the differences are a matter of degree. For example, VSH-selected bees may be more sensitive to the odors associated with *Varroa* parasitized brood as shown by Wagoner et al (2019) [44].

In 2024, there was no difference in mite infestation between the two lines of bees and very low mite growth over the season. One reason for the low mite levels could be because all colonies were treated in 2024 to equalize mites across colonies prior to challenge. The lack of mite growth could also be due to the smaller colony size, as all colonies in 2024 were confined to one brood box, whereas in 2023 they had two. Another reason could be due to the decreases in sample size. Over the course of the experiment, seven Pol-line colonies and seven Commercial colonies were lost due to queen failures, mostly supersedures. Queen failures are one cause of honey bee colony mortality [56,57]. Pesticides, parasites, heat-shock, and cold-shock can all negatively impact a queen's reproductive potential and increase the likelihood of queen failure [58–64]. The probability of supersedure increases with increasing level of virus in the queens [65], but we did not see any influence of viral load in worker bees on queen failure likelihood [66]. We cannot be certain of the causes for high rates of queen failure observed in 2024.

## Freeze-killed brood

We did not observe any significant relationships between the FKB assay and chalkbrood prevalence, chalkbrood severity, or mite loads. Most of the colonies in this study had high levels of hygienic behavior based on the FKB assay, and thus the low variation in FKB results may have obscured potential predictive relationships. It was not surprising that the two lines of bees scored similarly on the FKB assay, as bees selected for the VSH-trait (Pol-line) are known to be highly hygienic [21,67].

How quickly a colony can detect and remove experimentally killed brood (by FKB) does not always correspond with the colony's ability to detect and remove diseased brood [4]. Gilliam et al (1988) observed that there was a relationship between uncapping and removal of FKB and resistance to chalkbrood, but this relationship was not consistent across the 31 colonies tested [6]. Taber (1986) also noted similar exceptions to FKB and chalkbrood resistance [68]. Spivak and Gilliam (1993), found that some colonies challenged with chalkbrood became infected despite their high FKB removal

rates and hygienic tendencies [26]. Studies in Australia reported that hygienic behavior selection via FKB was not associated with chalkbrood resistance [27]. The FKB assay is a simple and easy screening assay to find hygienic colonies that may have behavioral resistance to pathogens and parasites, but after screening, it is important to subsequently challenge colonies with a pathogen or parasite to determine if they are behaviorally resistant [13,17].

## Conclusion

The findings from this study indicate that the Pol-line shows resistance to mites and resistance to the pathogen that causes chalkbrood disease. Challenging colonies selected for VSH and general hygiene with another pathogen like *Paenibacillus larvae* that causes American Foulbrood could help confirm these findings. On a practical level, this research supports that the use of VSH lines of bees in the beekeeping industry. Colonies with demonstrated resistance to parasites and pathogens (i.e., the Pol-line) would help to improve the health and resilience of colonies, as well as reduce economic costs for treatment.

It is worth exploring whether VSH and general hygiene are the same traits but expressed at varying degrees. For example, bees that are selected for VSH may have lower olfactory response thresholds to odors associated with any brood that is abnormal (dead, diseased, or mite-infested) compared to bees with general hygiene. Directly comparing the olfactory sensitivity of bees from colonies selected for VSH and bees selected by FKB could shed light on if these traits are genetically distinct or continuous. Further side-by-side behavioral observations of colonies and of individual bees from both lines could help inform and interpret future -omic studies on underlying mechanisms.

## Supporting information

**S1 Table. Summary of methods used in 2023 and 2024 and between locations.**
(DOCX)

**S2 Table. Sample sizes of pre-pupae pooled for detection of chalkbrood by RT-qPCR.** Number of pooled pre-pupae were restricted by how many symptomatic pre-pupae were found on the comb. In the of colony 24429, only 6 pre-pupae were amorphous and only 14 pre-pupae were acuminate within the comb.
(DOCX)

**S3 Table. Primer sequences utilized for screening of pathogens by RT-qPCR.**
(DOCX)

**S4 Table. Prevalence of chalkbrood mummies and early signs by queen type.** Binomial regression statistical results comparing chalkbrood mummy prevalence and early sign prevalence by queen type (POL-line vs. Commercial). At 4 days post challenge in Baton Rouge, Pol-line colonies had significantly (86.4%) lower odds of having chalkbrood mummies compared to Commercial colonies ($\beta = -1.99$, $p = 0.034$). wk = week, d = day.
(DOCX)

**S5 Table. Colony strength in 2023 and 2024 prior to challenge.** Mean frames (fr) of bees and brood for Pol-line (Pol) and Commercial (Comm) colonies. Wilcoxon tests demonstrate difference between Pol-line and Commercial colonies. **In 2023**: Pol-line had significantly fewer frames of bees at day 0 ($p = 0.03$), however, there was no difference in frames of brood or brood pattern. **In 2024**: Wilcoxon tests revealed no statistically significant differences between Pol-line and Commercial colonies.
(DOCX)

**S6 Table. Hygienic behavior did not predict chalkbrood prevalence or severity.** To investigate the relationship between Freeze Killed Brood (FKB) as a predictor of chalkbrood prevalence, a binomial regression model was used. A Firth Penalized Regression model was used if there was perfect separation of the variables. FKB and chalkbrood

severity were analyzed via a linear model or Spearman Rank Correlation if the data was non-normal. Wk = week. d = day. FKB = Freeze-killed brood.
(DOCX)

**S7 Table. Freeze-killed brood as a predictor of mite infestation.** Spearman rank correlations indicated that FKB is not a predictor of mites/100 adult bees. FKB = Freeze-killed brood.
(DOCX)

## Acknowledgments

Nelson Williams, University of Minnesota Bee Lab, Minneapolis, MN.
   Ben Ziegler, University of Minnesota Bee Lab, Minneapolis, MN.
   Dr. Sofia Nikulin, University of Minnesota Bee Lab, Minneapolis, MN.
   Dr. Declan Schroeder, University of Minnesota Schroeder Lab, Minneapolis, MN.
   Joseph McCarthy, USDA-ARS Honey Bee Breeding, Genetics, and Physiology Unit, Baton Rouge, LA.
   Mandy Frake, USDA-ARS Honey Bee Breeding, Genetics, and Physiology Unit, Baton Rouge, LA.
   Garrett Dodds, USDA-ARS Honey Bee Breeding, Genetics, and Physiology Unit, Baton Rouge, LA.
   Melissa Fontaine, USDA-ARS Honey Bee Breeding, Genetics, and Physiology Unit, Baton Rouge, LA.

## Author contributions

**Conceptualization:** Isabell Dyrbye-Wright, Elizabeth M. Walsh, Marla Spivak.

**Data curation:** Isabell Dyrbye-Wright, Michael Simone-Finstrom.

**Formal analysis:** Isabell Dyrbye-Wright.

**Funding acquisition:** Isabell Dyrbye-Wright, Michael Simone-Finstrom, Marla Spivak.

**Investigation:** Isabell Dyrbye-Wright, Michael Simone-Finstrom, Elizabeth M. Walsh, Marla Spivak.

**Methodology:** Isabell Dyrbye-Wright, Michael Simone-Finstrom, Elizabeth M. Walsh, Marla Spivak.

**Project administration:** Michael Simone-Finstrom, Elizabeth M. Walsh, Marla Spivak.

**Resources:** Isabell Dyrbye-Wright, Michael Simone-Finstrom, Elizabeth M. Walsh, Marla Spivak.

**Software:** Isabell Dyrbye-Wright.

**Supervision:** Michael Simone-Finstrom, Elizabeth M. Walsh, Marla Spivak.

**Validation:** Isabell Dyrbye-Wright, Michael Simone-Finstrom, Marla Spivak.

**Visualization:** Isabell Dyrbye-Wright.

**Writing – original draft:** Isabell Dyrbye-Wright.

**Writing – review & editing:** Isabell Dyrbye-Wright, Michael Simone-Finstrom, Elizabeth M. Walsh, Marla Spivak.

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
