## [Decision Letter · Decision Letter 0]

7 Jun 2025

PLOS ONE

Dear Dr. Dyrbye-Wright,

Thank you for submitting your manuscript to PLOS ONE. After careful consideration, we feel that it has merit but does not fully meet PLOS ONE’s publication criteria as it currently stands. Therefore, we invite you to submit a revised version of the manuscript that addresses the few minor points raised by Reviewer #2.

We look forward to receiving your revised manuscript.

Kind regards,

Wolfgang Blenau

Academic Editor

PLOS ONE

Journal Requirements:

3. Thank you for uploading your study's underlying data set. Unfortunately, the repository you have noted in your Data Availability statement does not qualify as an acceptable data repository according to PLOS's standards.

Reviewers' comments:

Reviewer's Responses to Questions

**Comments to the Author**

1. Is the manuscript technically sound, and do the data support the conclusions?

Reviewer #1: Yes

Reviewer #2: Yes

2. Has the statistical analysis been performed appropriately and rigorously?

Reviewer #1: Yes

Reviewer #2: I Don't Know

3. Have the authors made all data underlying the findings in their manuscript fully available?

Reviewer #1: Yes

Reviewer #2: Yes

4. Is the manuscript presented in an intelligible fashion and written in standard English?

Reviewer #1: Yes

Reviewer #2: Yes

Reviewer #1: This is an exceptionally well-done study comparing two different lines of selected bees against two different pathogens. On top of that, it is replicated across two widely-separated sites. I cannot find any problems with the study.

Reviewer #2: This study investigates whether honey bee colonies bred for Varroa Sensitive Hygiene (VSH) also exhibit resistance to the fungal brood pathogen Ascosphaera apis (chalkbrood). Colonies from a VSH-selected Pol-line were compared to commercial colonies selected for general hygienic behavior across two years and two locations. The authors conclude that both lines show similar levels of chalkbrood resistance

General Comments

The study addresses a timely and practical question in honey bee breeding and health. The experimental design is ambitious, spanning years and locations, but some methodological details are unclear in the manuscript, particularly around the statistical analyses. The manuscript would benefit from clearer explanations of how key statistical values were derived, inclusion of mixed effects where appropriate, and correction for multiple testing all if needed. The figures and tables are mostly clear, though some terminology and labeling could be improved for clarity.

Line-by-Line Comments

Line 42: The claim that commercial colonies were "not intentionally selected for Varroa resistance" seems unlikely. Many commercial beekeepers test for Varroa and may propagate from surviving or mite-tolerant colonies. This statement should be clarified or softened.

Line 57: The opening sentence is quite bold. Can you cite evidence or rephrase more cautiously?

Lines 203 and 209: It's unclear how these statistical values were derived. Please specify the model used and how comparisons were made.

Lines 226–228: Since the results begin at line 221 and continue into line 226, consider merging these paragraphs for improved flow.

Line 228: Again, the statistic referenced here is not clearly explained. What is being tested, and how?

Line 251 and Figure 6: This section would benefit from more rigorous statistical treatment. Given the multiple years, dates, and locations, you should consider using mixed-effects models and adjusting for multiple comparisons to reduce the risk of false positives.

Line 310: The relevance of the cited reference here is unclear. Consider explaining its significance or removing it.

Table 1 (line 546): By visual inspection, the difference between lines looks minimal. Your statistical model does not appear to include “location” as a fixed effect, which could obscure meaningful variation.

Table 2: With multiple years, dates, and variables, it appears you've run ANOVAs separately for each. While this may be adequate given the lack of strong effects, any significant results found this way may be spurious. Consider applying p-value correction for multiple testing and possibly mixed-effects modeling.

Figures 1–3: Consider relabeling the y-axes to just “Severity” for clarity and consistency.

**Do you want your identity to be public for this peer review?** For information about this choice, including consent withdrawal, please see our Privacy Policy

Reviewer #1: **Yes: ** Leonard Foster

Reviewer #2: No

---

## [Author Response · Author response to Decision Letter 1]

17 Jul 2025

Responses to Reviewers:

Line-by-Line Comments

Line 42: The claim that commercial colonies were "not intentionally selected for Varroa resistance" seems unlikely. Many commercial beekeepers test for Varroa and may propagate from surviving or mite-tolerant colonies. This statement should be clarified or softened.

Response: Thank you for pointing this out. We added “..in this study” at line 42 to clarify that the commercial colonies used in this study were not intentionally selected for varroa resistance, but rather for general hygienic behavior.

Line 57: The opening sentence is quite bold. Can you cite evidence or rephrase more cautiously?

Response: Good point, we softened the language to “Honey bee health is compromised from many stressors, including various pathogens and parasites” at line 57.

Lines 203 and 209: It's unclear how these statistical values were derived. Please specify the model used and how comparisons were made.

Response: Thank you for highlighting this. We added clarification in the methods section on line 278 to clarify that the model used at line 278 as follows: “As the data were non-normal, a Kruskal-Wallis test was used to assess if there were any differences in chalkbrood loads detected by real-time RT-qPCR between the early sign categories: acuminate, amorphous, and uncapped pre-pupae”. We also changed virus titers to chalkbrood titers at line 373 to increase understanding. The statistical model used to derive the values on line 405 is on line 285 of the statistics section: “All other variables (severity scores for chalkbrood mummies and early signs, hygienic behavior, and mite loads) were compared between queen types by analysis of variance (ANOVA). If the data were non-normal and did not meet heteroscedasticity requirements a Kruskal Wallis test was used in place of ANOVA”

Lines 226–228: Since the results begin at line 221 and continue into line 226, consider merging these paragraphs for improved flow.

Response: We appreciate you brining this to our attention. We merged the section into one paragraph at line 415 to improve flow. We also clarified that the highest severity scores were peak infection.

Line 228: Again, the statistic referenced here is not clearly explained. What is being tested, and how?

Response: Thank you for pointing this out. We added the explanation to the statistic method section on line 296: “To investigate how chalkbrood severity differed between 2023 and 2024 in Minnesota at 7 days and 14 days, a Kruskal-Wallis test was used as the data was non-normal and did not meet heteroscedasticity requirements.”

Line 251 and Figure 6: This section would benefit from more rigorous statistical treatment. Given the multiple years, dates, and locations, you should consider using mixed-effects models and adjusting for multiple comparisons to reduce the risk of false positives.

Response: Thank you for this insight. We adjusted the beginning of the statistic method section at line 274 to clarify that all models were run separately, largely because there were non-random differences in chalkbrood severity between the two locations and over the years due to methodology. Using mixed effect models would be another way to run these data. However, rather than using mixed effect models and adjusting for multiple comparison, we opted to run all of our models separately. We reiterate this in more detail to the comment made about Table 2.

Line 310: The relevance of the cited reference here is unclear. Consider explaining its significance or removing it.

Response: This is a salient point. We expanded on what -omics mean on line 558 (…”genomic, transcriptomic, and proteomic (-omic) studies”) to give more clarity to reference (37) which is Mondet et al 2020. Mondet et al 2020 is of great significance to this study as it is a meta-analysis of proteomic, genomic, and transcriptomic studies that investigated genes (QTLs and SNPs) that might influence hygienic behaviors such as VSH and general hygiene. The study found a lack of uniformity in genes influencing the VSH-trait and general hygiene trait. Our study was a field study (rather than an -omic study) that investigated how bees bred for the VSH-trait vs. general hygiene trait responded to chalkbrood disease. Our results give insight into how important field studies are in addition to -omic studies.

Table 1 (line 546): By visual inspection, the difference between lines looks minimal. Your statistical model does not appear to include “location” as a fixed effect, which could obscure meaningful variation.

Response: Thank you for bringing this to our attention. We added clarification at line 274 to reiterate that location is not needed as a fixed effect because data from locations were never pooled together, but kept sperate by location and year due to differences methodology.

Table 2: With multiple years, dates, and variables, it appears you've run ANOVAs separately for each. While this may be adequate given the lack of strong effects, any significant results found this way may be spurious. Consider applying p-value correction for multiple testing and possibly mixed-effects modeling.

Response: We added clarification in the statistics method section at line 274. We acknowledge that p-value correction for multiple testing and mixed effect modeling is another way to run these data, but we opt to keep our models separately to be consistent with the rest of the data analysis. This is largely because there were non-random differences in chalkbrood severity between the two locations and over the years. As Baton Rouge has a more humid climate, chalkbrood grows faster and colonies were setup following a different approach. Furthermore, our change from the pollen patty to the spore spray increased chalkbrood inoculation infectivity across years, along with the additional disease phenotype measured in 2024. So, we think it is best to fit each model for each group separately to mitigate risk of pooling assumptions.

Figures 1–3: Consider relabeling the y-axes to just “Severity” for clarity and consistency.

Response: Good idea, figures amended for consistency.

---

## [Editor Report · Decision Letter 1]

22 Jul 2025

Honey bees bred for Varroa Sensitive Hygiene trait demonstrate resistance to chalkbrood disease

PONE-D-25-23856R1

Dear Dr. Dyrbye-Wright,

We’re pleased to inform you that your manuscript has been judged scientifically suitable for publication and will be formally accepted for publication once it meets all outstanding technical requirements.

Kind regards,

Olav Rueppell

Academic Editor

PLOS ONE
---

## [Editor Report · Acceptance letter]

PONE-D-25-23856R1

PLOS ONE

Dear Dr. Dyrbye-Wright,

I'm pleased to inform you that your manuscript has been deemed suitable for publication in PLOS ONE. Congratulations! Your manuscript is now being handed over to our production team.

Kind regards,

on behalf of

Dr. Olav Rueppell

Academic Editor

PLOS ONE